


# 1 Brief Communication: Key papers of 20 years in Natural Hazards and

# 2 Earth System Sciences

Animesh K. Gain[1], Yves Bühler[2], Pascal Haegeli[3], Daniela Molinari[4], Mario Parise[5], David J. Peres[6], Joaquim G. Pinto[7], Kai
Schröter[8], Ricardo M. Trigo[9], María Carmen Llasat[10], Heidi Kreibich[8]
[1]Department of Economics, Ca' Foscari University of Venice, Cannaregio 873, 30121 Venice, Italy
[2]WSL Institute for Snow and Avalanche Research SLF, 7260 Davos Dorf, Switzerland
[3]School for Resource and Environmental Management, Simon Fraser University, Burnaby BC V5A 1S6, Canada
[4]Department of Civil and Environmental Engineering, Politecnico di Milano, Piazza Leonardo da Vinci 32, 20133 Milan, Italy
[5]Earth and Environmental Sciences Department, University Aldo Moro, Bari, 70126, Italy
[6]Department of Civil Engineering and Architecture, University of Catania, Catania, 95123, Italy
[7]Institute of Meteorology and Climate Research, Karlsruhe Institute of Technology, 76131 Karlsruhe, Germany
[8]GFZ German Research Centre for Geosciences, Section Hydrology, 14473 Potsdam, Germany
[9]Instituto Dom Luiz, Faculdade de Ciências da Universidade de Lisboa, Campo Grande, 1749-016, Lisboa, Portugal
[10] Department of Applied Physics, University of Barcelona, Barcelona, 08028, Spain
*Correspondence to*: Animesh K. Gain (animeshkumar.gain@unive.it)

**Abstract**
To mark the twentieth anniversary of Natural Hazards and Earth System Sciences (NHESS), an interdisciplinary and
international journal dedicated to the public discussion and open-access publication of high-quality studies and original
research on natural hazards and their consequences, we highlight eleven key publications covering major subject areas of
NHESS that stood out within the past 20 years. The selected articles represent excellent scientific contributions in the major
areas of natural hazards and risks and helped NHESS to become an exceptionally strong journal representing interdisciplinary
areas of natural hazards and risks. At its 20th anniversary, we are proud that NHESS is not only used by scientists to disseminate
research results and innovative novel ideas but also by practitioners and decision-makers to present effective solutions and
strategies for sustainable disaster risk reduction.

# 27 1 Introduction

Embracing a holistic earth system science approach, Natural Hazards and Earth System Sciences (NHESS) is an
interdisciplinary and international journal dedicated to the public discussion and open-access publication of high-quality
studies and original research on natural hazards and their consequences. NHESS serves a wide and diverse community of
research scientists, practitioners, and decision makers concerned with detection, monitoring, analyses and modelling of natural




hazards and risk. This includes the design and implementation of risk management and adaptation strategies, considering
economical, societal, and educational aspects. NHESS started its journey in the year 2001, when the world experienced the
Gujarat earthquake with a death toll more than 15000 people. Since its inception in 2001, NHESS has so far published 3436
articles until October 08, 2021. Initially (2001-2002), the number of published issues per year was only four, while this number
increased to six per year during 2003-2009, and further increased to 12 per year since 2010. Currently, there are ten subject
areas in the journal: atmospheric, meteorological, and climatological hazards; sea, ocean, and coastal hazards; hydrological
hazards; landslides and debris flow hazards; earthquake hazards; volcanic hazards; other hazards (e.g. glacial and snow
hazards, karst, wildfire hazards, and medical geohazards); databases, GIS, remote sensing, early warning systems, and
monitoring technologies; risk assessment, mitigation and adaptation strategies, socio-economic and management aspects; and
dissemination, education, outreach, and teaching. The word cloud of the selected articles are shown in Figure 1.

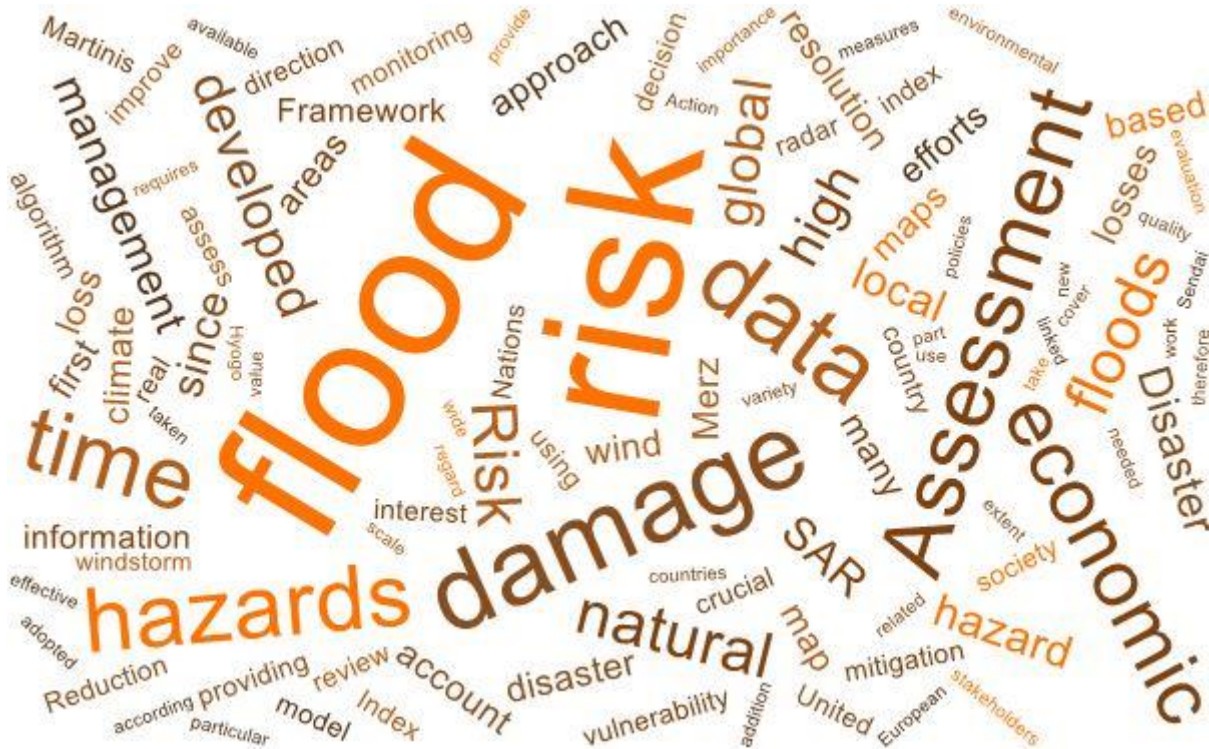


**Figure 1: Word cloud of selected articles published in Natural Hazards and Earth System Sciences**

To mark the twentieth anniversary of NHESS, we showcase  eleven key publications covering major subject areas of NHESS
that stood out within the past 20 years. These publications were chosen taking into account their relevance and impact to the
wider NHESS audience, often translated into high number of citations. Some of the hazards highlighted in our overview are
closely related to weather driven mechanisms that can be amplified by the ongoing climate change to various degrees, as
mentioned in the latest IPCC Assessment Report.




For example, the selected article on 'Assessment of economic flood damage' (Merz et al., 2010) is one of the highly cited

interdisciplinary articles (637 citations, based on Scopus search dated October 5, 2021), which covers multiple topics such as

*hydrological hazards*, *meteorological hazards*, and *risk assessment, mitigation and adaptation strategies, socioeconomic and*

*management aspects*. The development of multi-hazards disaster risk index by Peduzzi et al. (2009), a paper with more than

278 citations, was one of the initial contributions on was one of the initial contributions on quantitative assessments of risks

globally, within the topic of *risk assessment, mitigation and adaptation strategies, socioeconomic and management aspects.*

In the topic area of *remote sensing*, Martinis et al. (2009) developed one of the first algorithms for near-real-time

flood detection by using high-resolution Synthetic Aperture Radar (SAR) satellite data. Klawa and Ulbrich (2003) developed

one of the very first simple but effective storm loss models in the area of *atmospheric, meteorological and climatological*

*hazards*. In the area of *landslides and debris flow hazards*, Bogaard and Greco (2018) developed a conceptual model for

regional landslide hazard assessment based on physical process understanding and empirical data. Within the topic of other

hazards, we selected two very relevant recent studies: (i) predicting fire-weather index and its capacity by Di Giuseppe et al.

(2020) based on the ensemble forecast system of the European Centre for Medium-Range Weather Forecasts; and (ii) spatial

consistency and bias in public avalanche forecasts by Techel et al. (2018). In the area of *volcanic hazards* and *dissemination,*

*education, outreach and teaching*, we highlight an interesting article on the innovative use of video games for volcanic hazard

education and communication by Mani et al. (2016). Considering the importance of social psychology of seismic hazard

adjustment in household level, we selected the contribution of Solberg et al. (2010). In the area of *sea, ocean and coastal*

*hazards*, the contribution by Monserrat et al. (2006) on similarities and differences between seismic and meteorological

tsunamis was innovative. Using multiple drought indices, the important contribution by Sousa et al. (2011) helped analysing

the spatial and temporal evolution of drought conditions in the Mediterranean during the 20th century. The papers thus cover

all the subtopics contemplated in the division on Natural Hazards including dissemination, education, outreach and teaching.

## 2 Contributions of selected articles

### 2.1 Economic damage assessment of floods

Assessment of natural hazards covers a wide range of disciplines, representing an issue, which is of sure interest to society,

given the casualties and economic losses registered annually. At this regard, the review article "Assessment of economic flood

damage" by Merz et al. (2010) is a remarkable contribution, dedicated to assessment of the damage related to floods. This is a

topic, which is gaining increasing interest from many stakeholders, since it is a crucial element in the policies of flood risk

management. In times when we have to face problems linked to climate changes, and adapt our way of lives to mitigate the

risks related to natural hazards, such an approach is definitely of primary importance. A variety of flood maps exists in the

different countries of Europe, according to national laws. The flood directive of the European Union (EU) also requires

Member States to map the flood extent and to assess the assets and humans at risk and to take adequate and coordinated



measures to reduce the flood risk (EC, 2007). In theory, flood risk maps include an assessment of the possible economic losses
on society. However, this is rarely effective as many "risk maps" in practice do not cover all needed elements, and should
more correctly be defined as hazard maps. The incorrect use of the terms creates therefore a serious drawback in the overall
management of the risk. Typically, assessment of the hazard plays a much prominent part with respect to that regarding the
damage, and this results in a mismatch in the quality of the available models and datasets for evaluating the economic damage.
Therefore, the thorough review of methods for the assessment of economic flood damage provided by Merz et al. (2010) was
and is still of high value for both practitioners and scientists. However, we should also mention that many new approaches
have been developed in the meantime, since the review was published in 2010.
Even though the article by Merz et al. (2010) is focused on flood damage assessment, issues as the risk-based evaluation of
mitigation measures, and the methodological aspects of damage estimation are valid for other natural hazards, too. This still
increases its value, and the positive effect on the scientific community. A crucial point, worth of further work and of particular
interest to NHESS readers, is the statement that flood risk cannot be managed alone: in areas affected by many geological
hazards, these should all be considered in the policy of risk mitigation, according to magnitude of the phenomena and historical
records of their effects. Introducing economic issues, such as the considerations about stock and flow values, in an article
dealing with natural hazards is certainly part of a forward-looking vision, aimed at providing useful tools to decision-makers
in order to develop the most proper actions for flood risk management. It has to be pointed out, however, that more efforts are
needed in this direction: for instance, in addition to economic flood damage, here taken into account, the adverse social,
psychological, political and environmental consequences should be examined, in order to gain a comprehensive picture of the
damage.
**2.2 Disaster risk index**
Efforts to assess and map natural hazard risk at the global scale have been ongoing since the mid-2000s in order to provide
science-based information for disaster risk management. The global disaster risk management approach was formally adopted
by international policies such as the Hyogo Framework For Action and the Sendai Framework for Disaster Risk Reduction.
The priority two of the Hyogo Framework For Action states that "The starting point for reducing disaster risk […] lies in the
knowledge of the hazards and the […] vulnerabilities to disasters […] followed by action taken on the basis of that knowledge".
The priority was further considered by the following Sendai Framework for Disaster Risk Reduction (UNISDR, 2015). Using
the definition by the United Nations Development Programme in 2004, the Disaster Risk Index (DRI) (Peduzzi et al., 2009)
was the first attempt to produce a global, quantitative approach to assessing risk due to multiple hazards. By exploring the
relationship between human losses and socio-economic and environmental variables for a variety of hazards (i.e., cyclones,
droughts, earthquakes and floods), Peduzzi et al. (2009) provided the first statistical evidence of the links between vulnerability
to natural hazards and levels of development at the global scale (i.e. country by country). The study helped in supporting aid
organisations and governments through comparing countries across risk levels and hazard types, with an aim to make decision
on risk mitigation strategies in time. In fact, since 2009, the index has been adopted in the Global Assessment Reports (GAR)



of the United Nations Office for Disaster Risk Reduction, leading in 2017 to the publications of the GAR Risk Atlas (UNDRR,
2017) providing globally multi-hazard risk metrics. The significance of the DRI is further proved by the numerous researches
that were carried out in the same direction, providing alternative indexes to assess risk, at the global or at the local level.
Among them, the Index For Risk Management (INFORM) was developed by the European Joint Research Centre and is
published on the homonymous webGIS platform twice a year. Additional efforts were also devoted to the inclusion of climate
change impact in the evaluation. Indeed, vulnerability of a country is considered as a key criterion also to decide on climate
adaptation funding. The World Risk Index (WRI) can be quoted as an example in this direction.  The WRI was developed by
the United Nations University – Institute for Environment and Human Security (Welle and Birkmann, 2015), and is now
published by the Institute for International Law of Peace and Armed Conflict of Ruhr-University Bochum. It allows to take
into account of climate change vulnerability and adaptive/coping capacity.

**2.3 Real-time flood detection using remote sensing**

When floods strike, emergency response and disaster relief need rapid information of the situation on the ground. In this
context, technological advancements open new possibilities for supporting crisis intervention. The provision of inundation
extent from satellites in near-real-time is one such success story. Situational awareness during floods requires reliable
information with a high spatial resolution to locate worst-hit areas and aid decision-making concerning the identification of
target areas for distributing resources. Satellite data improve our capability to detect, map and monitor river floods and their
impacts at local and global scales. For flood monitoring, it is advantageous and effective to utilize active sensors. In particular,
radar is suitable as it penetrates rain and cloud cover, which are issues in flood-hit locations. In this regard, very high-resolution
Synthetic Aperture Radar (SAR) data show enormous potential to improve the reliability of flood mapping. However, there is
usually only limited time and personnel available during emergencies to understand and process geospatial data into
meaningful products. Automatic processing algorithms are crucial to reducing the time lag between data acquisition and flood
map dissemination. The algorithm developed by Martinis et al. (2009) is one of the first algorithms that enable the completely
unsupervised detection of inundated areas from very high-resolution SAR data in near-real-time. It builds on a split-based
threshold for extracting low backscattering from open flood surfaces in SAR data in a fully automatic and time-efficient
manner. The segmentation of the radar scene and the context-sensitive threshold, in addition to the radar reflectivity,
incorporate topological information into the classification. As the authors demonstrate, this enhances the quality of the
outcomes. Notably, the algorithm does not require training data and is very suitable for applications even when the acquisition
of ground-truth data is not feasible. With this development, the authors leverage very high-resolution SAR data for near real-
time flood mapping in operational flood monitoring systems and improve our emergency response capabilities (Martinis et al.,
2015; Matgen et al., 2011). Today, numerous flood monitoring services are in operation using SAR data with unsupervised
classification (Schumann et al., 2018). The algorithm developed by Martinis et al. (2009) is a cornerstone in this advancement.





**2.4 Assessment of storm losses**

The quantification and forecasting of impacts associated with the occurrence of natural hazards like windstorms or floods is of major importance for society and stakeholders (e.g., Merz et al. (2020)). One of the first efforts to provide a simple but physically based quantification of windstorm associated damage to buildings and infrastructure was the seminal work of Klawa and Ulbrich (2003). The authors considered daily maximum wind gusts from German weather stations, which were scaled by the local 98th percentile to account for local wind conditions and determine the area where damage potentially occurred (windstorm footprint). The scaled wind gusts exceeding the $98^{th}$ percentile are cubed ($V^3$) to account for the wind's destructive power, and are weighted with the population density (a proxy for the local insured property). The authors found high correlations between their loss model and the loss data from the German Insurance Association.

The loss model by Klawa and Ulbrich (2003) has since proved to be a highly efficient and widely applicable approach, becoming a very popular and easy-to-use socio-economic loss model for insurance applications, and leading to a wide number of further developments and follow up studies. For example, Leckebusch et al. (2008) developed the concept of the storm severity index (SSI) further and considered "wind tracking", where the windstorm footprints for a certain time frame are linked together in space and time. Pinto et al. (2012) explored the differences between the extremeness of windstorms when considering purely meteorological versus population-weighted impacts.

The method has been applied to Reanalysis datasets and both global and regional climate model data, permitting the quantification of the windstorm risk in Europe and elsewhere for recent and future decades (e.g., Pinto et al. (2012); Leckebusch et al. (2008)). Recently, Pantillon et al. (2017) provided evidence that the impact of European windstorms is predictable with a certain level of confidence with a lead time of 2-4 days using 20 years of European Center for Medium-Range Forecasts (ECMWF) ensemble forecast data. This demonstrates the ability to assess storm damage, issue extreme weather warnings in a timely manner, and respond appropriately to avoid major damage and disruption.

**2.5 Landslide triggering thresholds**

Landslides triggered by rainfall cause damage and casualties worldwide (Froude and Petley, 2018). The implementation of landslide early warning systems is one of the most important measures for protecting populations at risk. A fundamental step for setting up an early warning system is the identification of the relationship between the precursors and landslide occurrence (Segoni et al., 2018). A large number of papers have treated this problem by attempting to derive thresholds expressed in the form of a power-law between rainfall event duration and mean intensity or event rainfall (the total rainfall depth accumulated over rainfall event duration), inspired by the pioneering paper by Caine (1980). Not many researchers have questioned this method for decades.

With their invited perspective, Bogaard and Greco (2018) discussed some theoretical reasons to move beyond this traditional approach. They stress that thresholds based only on rainfall event characteristics may not sufficiently reflect the hydrological processes occurring along slopes. In particular, intensity-duration thresholds do not allow to explicitly take into account the



fact that the triggering rainfall event may be just the final "push" (trigger) after a longer wet period that predisposed the slope
to fail (cause). They argued then that the cause-trigger concept may be better represented by hydro-meteorological thresholds.
The term hydro-meteorological refers to the fact that these types of thresholds should combine a meteorological variable
(rainfall depth) with a hydrological one, reflecting the water storage at the catchment or local scale.
Water stored in the unsaturated zone, is however a variable that is more difficult to measure with respect to precipitation. On
the other hand, soil moisture information is increasingly becoming available, thanks to remote sensing missions. Reanalysis
datasets have attracted the attention by researchers in this field as well. Within this context of an increase of availability of soil
moisture information, the perspective paper soon stimulated an increasing number of scholars (i.e., cited 90 times in last three
years) to investigate the use of the hydro-meteorological approach to improve the performances of empirical thresholds
indicating landslide triggering conditions (Mirus et al., 2018; Marino et al., 2020; Reder and Rianna, 2021). The way through
this improvement remains however quite challenging. Soil moisture presents high spatial and temporal variability, and remote
sensing products – as well as reanalysis ones – are available only at coarse temporal/spatial resolutions; comparisons with in
situ measurements have shown that accuracy issues may be present as well. Notwithstanding such obstacles to deal with, the
invited perspective is stimulating scholars to move beyond an approach that remained nearly unquestioned for many years.
**2.6 The prediction of Fire-weather Indices**
Even if a commonly accepted definition is still lacking, it is becoming widely recognized that we are currently living in the
Anthropocene epoch. The impact on the makeup of our planet's atmosphere, as well as on the disruption of many biomes and
ecosystems are part of the Anthropocene fingerprint. In this context, it is important to stress that the three critical components
that control the triggering and spread of wildfires (i.e. ignitions, fuels and weather/climate) are, to a large extent, influenced
by human activities. Thus, the higher concentration of greenhouse gases produced by mankind is already increasing
significantly the likelihood of heatwaves (Fischer and Knutti, 2015) that are often linked to more intense and prolonged fire
seasons (Ruffault et al., 2020). Additionally, in many semi-arid areas of the globe the increasing temperatures coupled with a
decrease in precipitation are aggravating the dryness of fuels (Abatzoglou and Williams, 2016).
Besides destroying property worth billions of Euros, wildfires are still capable of impinging a disconcerting large number of
human fatalities, even in some of the most highly developed regions of the world, (e.g Portugal 2017, California and Greece
2018, Australia 2020). Prediction of many weather driven natural hazards (e.g. heatwaves, floods or tropical cyclones) reached
a fairly mature standard, however, the forecast of wildfire prone conditions still lags behind with fire danger indicators mostly
relying on environmental monitoring. In 2020, a study led by Francesca Di Giuseppe (Di Giuseppe et al., 2020) published in
NHESS suggested extending fire danger warnings with the use of the most advanced weather forecast model available, i.e.
the European Centre for Medium-Range Weather (ECMWF) models. By systematically evaluating the ECMWF ensemble
forecast system performance to reproduce fire weather index (FWI) from observing stations at the global scale, the authors
demonstrate the capacity of this ensemble approach to be reasonably accurate up to 10 days ahead, especially for some of
the largest fires that took place in 2017, namely in Chile and Portugal. Their results confirm that early warning could be





extended by up to 1–2 weeks by using advanced numerical weather models, allowing for better coordination of resource-
sharing and mobilization within and across countries (Di Giuseppe et al., 2020).

### 2.7 Avalanche forecasting

Since the inception of the journal NHESS, more than 80 avalanche research articles have been published covering a wide range
of topics including terrain mapping, hazard and risk assessment approaches, developments in avalanche runout models,
avalanche-forest interactions, assessments of risk mitigation approaches and others. Of the many excellent contributions, we
would like to highlight the paper of Techel et al. (2018), who examined the spatial consistency and bias in avalanche forecasts
across the European Alps. While globally the largest number of avalanche fatalities are caused by catastrophic avalanches
hitting villages or infrastructure in mountain ranges such as the Himalayas, more than 90% of avalanche deaths in western
countries involve backcountry recreationists who voluntarily expose themselves to avalanche hazard. For this user group,
avalanche forecasts published by local, regional or national avalanche warning services are a critical source of information for
developing an informed understanding of the existing conditions and deciding when, where and how to recreate in avalanche
terrain. Despite substantial scientific advances in our understanding of the factors affecting avalanche hazard and our ability
to predict it, the compilation of avalanche forecasts from a variety of different data sources still relies heavily on the personal
experience and judgment of avalanche forecasters, which makes it susceptible to inconsistencies and human biases.
Focusing on the avalanche danger ratings, a prominent component of avalanche forecasts, Techel et al. (2018) show that there
are considerable inconsistencies among the published ratings in the European Alps, and that the largest differences are mainly
found along national or agency boundaries and less between climatological or topographic regions where one would expect
them based on physical processes. These regional discrepancies make it challenging for backcountry users travelling across
forecast regions to properly understand the published ratings and apply them in a consistent way. In addition, these
inconsistencies can negatively affect the credibility of avalanche forecasts and lead to judicial problems in the case of avalanche
accidents. While experienced forecasters were aware of this challenge, the innovative analysis approach developed by Frank
Techel and his team was the first to explicitly quantify the issue in a way that circumvents the inherent challenges associated
with validating danger ratings. The resulting insights have played an important role in initiating informed conversations about
differences in avalanche forecasting practices and creating a meaningful foundation for evidence-based improvements in the
future.

### 2.8 Video game as hazard education and communication

In 2015, the United Nations formalised the Sendai Framework for Disaster Risk Reduction 2015–2030, which identified the
need for participating countries to "strengthen public education and awareness in disaster risk reduction", specifically
promoting the use of social media and community mobilisation campaigns and encouraging the education of all at-risk
communities (Unisdr, 2015). Considering the importance of science communication for the natural hazards, the *dissemination,*
*education, outreach, and teaching* is considered as one of the key subject for NHESS. However, this is less explored area in





natural hazards research.. 'Using video games for volcanic hazard education and communication' by Mani et al. (2016) is one
of the very few studies which contributes in this direction. They developed a video game for St. Vincent's Volcano in the
eastern Caribbean island with an aim to enhance residents' education and communication of potential future volcanic hazards.
The findings suggest that serious games have the potential to be effective tools in volcano education for both traditional (school
students) and non-traditional (i.e., adults) stakeholder groups. Though serious games, therefore is a promising communication
and educational technique, this approach faces a number of challenges such as expensive and time consuming processes of
game development. The study by Mani and his colleagues (Mani et al., 2016) offers exciting opportunities to build knowledge
and resilience among a diverse range of social groups within at-risk communities.

**2.9 The psychological factors shaping human adjustments of seismic hazards**

The risk reduction efforts of natural hazards including seismic hazards are at the forefront of discussions on contemporary
global forums such as the United Nations (UN) Sustainable Development Goals (SDGs) and the Sendai Framework for
Disaster Risk Reduction (SFDRR) (Rahman and Fang, 2019). Besides structural measures, non-structural measures including
emotional and socio-cultural factors play a key role in people's risk-related behavior for disaster risk reduction (Mohibbullah
et al., 2021). As people tend to be guided more strongly by their emotional reactions than by scientific or logical approach,
psychological adaptation to disasters is an interesting area of research. Given the importance, Solberg et al. (2010) reviewed
the psychological factors that shape human adjustments to seismic risk. This is one of the very few studies that synthesise the
major findings from the 40 years of the international literature on the psychological adjustments of seismic hazards including
the normative beliefs of earthquake protection responsibility and trust among key stakeholders of seismic risks (e.g.,
management authorities and local people). They also analyse the importance of seismic adjustment attributes such as beliefs
about efficacy, control and fate. The findings suggest that the consideration of norms, trust, power and identity play a key role
in seismic hazards adjustment. The article by Solberg et al. (2010) stimulated interesting discussion and further development
on psychological and behavioural adjustment of seismic hazard.

**2.10 Meteorological tsunamis**

Meteorological tsunamis (or simply known as meteotsunamis) are typically recognized as long ocean waves, which have the
same frequencies and spatial scales as tsunami waves of seismic origin, but produced by atmospheric processes. They are
triggered by extreme weather events atmospheric conditions at the ground or mid-troposphere including severe thunderstorms,
squalls (a sudden violent gust of wind or localized storm, especially one bringing rain, snow, or sleet), storm fronts, hurricanes
or instable intense mid-troposphere jets generating atmospheric gravity waves, generated through the rapid changes in
barometric pressure, (a few hectopascals over a few minutes) or wind  . The similarity between atmospherically generated
"meteotsunamis" and seismically generated tsunamis is strong enough that it can be difficult to distinguish one from the other.
The article by Monserrat et al. (2006) is one of the very few studies that describes the hazardous phenomena of meteotsunamis





in the World Ocean to show the similarities and differences with seismic tsunamis. Analysing several cases, Monserrat and his
team found that both tsunamis and meteotsunamis have the same periods, same spatial scales, similar physical properties and
affect the coast in a comparably destructive way. In addition, some specific features of meteotsunamis such as the coupling
between the moving disturbance and the surface ocean waves make them akin to landslide-generated tsunamis. Monserrat et
al. (2006) found that the major difference between the tsunamis and Meteotsunamis is associated with the specific properties
(mainly the resonant factors) of corresponding sources. During resonance of the ocean driven by atmospheric forcing, the
atmospheric disturbance propagating over the ocean surface is able to generate significant long ocean waves by continuously
pumping energy into these waves. This contrasts to seismic tsunamis that can have globally destructive effects without any
resonant factor. However, the Meteotsunamis are always local and much less energetic than seismic tsunamis. The destructive
meteotsunamis are always the result of a combination of several resonant factors such as Proudman, Greenspan, shelf, harbour.
As the probability of occurrence for such a combination is very low, the destructive meteotsunamis are infrequent and observed
only at some specific locations in the ocean.

## 2.11 Drier conditions in Mediterranean regions

The Mediterranean Region is considered a hot-spot of climate change. This qualification is supported by different natural and
socioeconomic reasons, being one of them its impact over hydrometeorological hazards, specifically, droughts.  Despite the
high uncertainty associated to the application of climatic models over the rainfall in this region, there is a high confidence on
the drought risk increase (Medecc, 2020), mainly due to precipitation reduction, a negative trend in moisture availability,  and
warming-enhanced evaporation. In a region where, in average, more than 65% of the freshwater is for agriculture near a 30%
is for the direct use of water by the population, and the remaining 5% is for industry, energy and tourism, droughts increase
implies that water related intersectoral conflicts are likely to be exacerbated.  Even more so if we consider that in 2025 about
530 million people will live in the Mediterranean, and that the increase in temperature will lead to an increase in irrigation
needs from 4 to 18% (Medecc, 2020). Although today there are already numerous studies at local and regional scale on the
observed spatial and temporal evolution of drought conditions, the paper by Sousa et al. (2011) updated the state of the art and
provided a robust and complete analysis of these conditions at Mediterranean scale during the 20th century.
Droughts constitute a complex and difficult risk to evaluate, so it is usual to define indices to estimate their onset, duration and
intensity. Sousa et al. (2011) applied the Palmer Drought Severity Index (PDSI) adapted to Europe (scPDSI) by the Climatic
Research Unit. The scPDSI is based on the water budget for a certain period estimated from precipitation, temperature and soil
characteristics and self-calibrated from local data. This index was applied to the Mediterranean Region and to four selected
sub-regions, homogeneous in terms of drought characteristics and socio-economic relevance, for the period 1900-2000.  After
a robust analysis the scPDSI showed a clear trend towards drier conditions in most Mediterranean Region. This index
reproduced well the strong decadal and inter-annual variability between subregions along all the century and showed how the
drought period recorded during the 1940s was extended from Iberia until the Balkans Region.  Having in mind that determined





synoptic patterns favours the deficit of precipitation and previous literature, and after analysing different major potential teleconnections, authors selected the North Atlantic Oscillation (NAO) and the Scandinavian index as the most representative for this region. The paper revealed the link between dry periods estimated by scPDSI and the positive phase of the NAO during winter and subsequent climatic seasons over the western Mediterranean, while the Scandinavian index presented a less homogeneous but significant pattern between winter and summer over central Mediterranean. Those teleconnections joined to the influence of the sea surface temperature (SST) anomalies allowed the creation of a stepwise regression model that was able to forecast summer drought conditions six months in advance and was capable of reproducing the observed scPDSI time series fairly well. Although it is a simple algorithm it provides a useful approach to seasonal forecasting of droughts, that can be very useful in a panorama characterized by an increase in dry periods.

## 3 Conclusion

The above articles represent excellent scientific contributions in the major subject areas of natural hazards and risks and helped NHESS to become an exceptionally strong journal representing interdisciplinary areas of natural hazards and risks. Pioneered in the open access model, NHESS is not only advancing scientific contributions and original research on broader areas of natural hazards and their consequences, but also the journal is dedicated to the public discussion engaging multiple stakeholders of natural hazards and risk communities. At its 20th anniversary, we are proud that NHESS is not only used by scientists to disseminate research results and innovative novel ideas but also by practitioners and decision-makers to present effective solutions and strategies for sustainable disaster risk reduction.

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
