# Peer review of "Brief Communication: Key papers of 20 years in Natural Hazards and"

_Natural Hazards and Earth System Sciences, 2021_

## Author Response (AR1)

**Response to the Reviewers Comments**

**Manuscript Number:** NHESS-2021-321

**Paper Title:** Brief Communication: Key papers of 20 years in Natural Hazards and Earth System Sciences

**Authors:** Animesh K. Gain[1], Yves Bühler[2], Pascal Haegeli[3], Daniela Molinari[4], Mario Parise[5], David J. Peres[6], Joaquim G. Pinto[7], Kai Schröter[8], Ricardo M. Trigo[9], María Carmen Llasat[10], Heidi Kreibich[8]

*Editor:*
The Editor has decided that minor revisions are necessary before the manuscript can be accepted.

**Response to *Editor***
**We are very pleased about the comments and appreciate the useful suggestions by the two reviewers Thom Bogaard and Francesca Di Giuseppe. We believe that they have substantially contributed to the improvement of the manuscript and consequently to the quality of this paper. In the revised manuscript, we have carefully considered the reviewer comments**. **Our detailed response to the comments of the reviewers are mentioned below.**

**Reviewer 1**

**Reviewer 1: General Comment 1**
I think the authors could elaborate on the process of the selection of the 11 papers. L 46-47 are a bit too concise to my taste. The authors do not need to justify the choices but a bit more description of the process would be nice. (See for example the paper of Bloschl et al 2019 on the 23 unsolved problem in hydrology. Also here, the process how the 23 questions were selected was described in detail). It seems the editorial board selected the key papers per subject field of NHESS and then the amount roughly based on the number of papers published in that topic. Or maybe there was an internal pre-selection and then a group discussion. Anyhow, a bit more clarity will help the readers.

**Response to *General Comment 1***
We appreciate the suggestion and accordingly we revise the introduction section specifically to detail the description of the process of the selection of the 11 papers. Please see our revision on lines 47-58 of the revised manuscript.

**Reviewer 1: General Comment 2**
Second, NHESS is incredibly broad and indeed publishes state-of-the-art science but also societal impact, outreach and education. This could be stressed more in the introduction (as it seems to have been a criteria for key paper selection as well), for example with a figure. I think the word cloud be replaced by a chord figure more emphasizing the interconnectivity and multi-discipline character of NHESS.

**Response to *General Comment 2***
As suggested by the reviewer, we now clearly describe the broader areas of NHESS. In addition, we replace the word cloud. As suggested by the second reviewer, the revised word cloud (Figure 1) is now based on titles of the published articles of the last 20 years in NHESS, instead of just on the selected 11 papers. The word cloud now represents keywords that have been used most in the titles of the journal articles.

**Reviewer 1:** Minor suggestions:

L47: some. Replace by exact number: 5 of the 11 papers etc..

L70-71: I think this sentence explains a lot and should be in the first paragraph of the introduction and in the abstract.

L320: Is NHESS really an Open-access discussion platform for broader audience? Of course all could take part, but in reality, in my perception, the open discussion is still very much peer-science based and not so much a multi-stakeholder discussion platform.

L321-323: I would not mind if this proudness is explicitly mentioned in abstract and introduction as well, because this is exactly what NHESS is doing very well

Blöschl G, et al. (2019): Twenty-three Unsolved Problems in Hydrology (UPH) – a community perspective, Hydrological Sciences Journal, DOI: 10.1080/02626667.2019.1620507

**Response to _Minor comments_**
We completely agree with the suggestions and revised the manuscript accordingly.

_Reviewer 2_

**Reviewer 2: General Comment 1**

The paper looks back at the first 20 years of NHESS activity and to the  achievements in the scientific exploration of natural hazard  understanding and prediction which the journal contributed to.

 The paper has an introduction summarising the topics covered by NHESS  and highlights its growth into the scientific community. Then it goes on to identify 11 papers to represent the vast inter disciplinal nature of the journal. The chosen 11 papers are summarised in more details and, given the different aspects they touch, they collectively provide a very nice overview of how natural hazard science have evolved during this time span.

I really enjoyed reading the summary of the key papers and I think so will do the readers. I therefore support this initiative and I have only one main suggestion for the introduction section an few minor comments/corrections for the second part.

I think the world cloud pic, that is only drawn from the key eleven paper, is a bit reductive of the broad spectrum of articles published in NHESS. I would suggest to either use the title of all the published articles in these first 20 years, or, if this is too laborious, maybe add a table which reports the number of papers that were produced in each of the 10 thematic subjects covered by the journal. It is also interesting to understand changes over time. From the discussion it emerges

that topics related to the perception of natural risk and human distress has only relatively recently been addressed by the NHESS community. It would be very interesting to see if there is an emergence in topics related to societal impact, outreach and education in the last period.

**Response to *General Comment 2**

We appreciate the comments and suggestions. We would like to thank the reviewer for the suggestion of replacing our initial word cloud. As suggested by the reviewer, we revise the word cloud (Figure 1) which is now based on the titles of all published articles of the last 20 years in NHESS, instead of just on the selected 11 papers. The word cloud now represents keywords that have been used most in the titles of the journal articles.

**Reviewer 2: Minor suggestions**

L25: remove innovative

L 49 IPCC Assessment Report: add citation

L 55 "was one of the initial contributions" repeated sentence

L62 : "and its capacity..." something is missing here

L67 adjustment at household level

L74-75: This sentence should be reworded as it is not very clear

L 78 way of living

L 79 remove "definitely". Later on, I think you mean "inundation map" not "flood maps"

L88 "However we should also mention" I would attach to the other sentence and say "so much so that many new approaches have been developed ...." The sentence now reads almost as a caveat and shouldn't.

L 247  I  don't like "serious game" maybe educational game ?

L 270 "squalls" I think should be "squall lines"

L271: "generated through" --> for

**Response to *Minor comments**
We completely agree with the suggestions and revised the manuscript accordingly.